

# Quantifying the impact of meteorological uncertainty on emission estimates and volcanic ash forecasts of the Raikoke 2019 eruption

Natalie J. Harvey[1], Helen F. Dacre[1], Cameron Saint[2], Andrew Prata[3], Helen N. Webster[2,4], and Roy G. Grainger[5]

[1]Department of Meteorology, University of Reading, Earley Gate, Reading RG6 6ET, UK
[2]Met Office, FitzRoy Road, Exeter EX1 3PB, UK
[3]Sub-Department of Atmospheric, Oceanic and Planetary Physics, University of Oxford, Oxford OX1 3PU, UK
[4]College of Engineering, Mathematics and Physical Sciences, University of Exeter, Exeter EX4 4QF, UK
[5]COMET, Sub-Department of Atmospheric, Oceanic and Planetary Physics, University of Oxford, Oxford OX1 3PU, UK

**Correspondence:** Natalie Harvey (n.j.harvey@reading.ac.uk)

**Abstract.** Due to the remote location of many volcanoes, there is large uncertainty in the timing, amount and vertical distribution of volcanic ash released when they erupt. One approach to determine these properties is to combine prior estimates with satellite retrievals and simulations from atmospheric dispersion models to create posterior emissions estimates constrained by both the observations and the prior estimates using a technique known as source inversion. However, the results are dependent not only on the accuracy of the prior assumptions, the atmospheric dispersion model and the observations used but also the accuracy of the meteorological data used in the dispersion simulations. In this study we advance the source inversion approach by using an ensemble of meteorological data to represent the uncertainty in the meteorological data and apply it to the 2019 eruption of Raikoke. This provides confidence in the posterior emission estimates and associated dispersion simulations that are used to produce ash forecasts. Prior mean estimates of fine volcanic ash emissions for the Raikoke eruption based on plume height observations are more than 15 times higher than any of the mean posterior ensemble estimates. In addition, the posterior estimates have a different vertical distribution with 27-44% of ash being emitted into the stratosphere compared to 8% in the mean prior estimate. This has consequences for the long-range transport of ash as deposition to the surface from this region of the atmosphere happens over long time-scales. The posterior ensemble spread represents uncertainty in the inversion estimate of the ash emissions. For the first 48 hours following the eruption, the prior ash column loadings lie outside an estimate of the error associated with a set of independent satellite retrievals whereas the posterior ensemble column loadings do not. Applying a risk-based methodology to an ensemble of dispersion simulations using the posterior emissions shows that the area deemed to be highest risk to aviation, based on the fraction of ensemble members exceeding predefined ash concentration thresholds, is reduced by 51% compared to estimates using an ensemble of dispersion simulations using the prior emissions with ensemble meteorology. If source inversion had been used following the eruption of Raikoke it would have had the potential to significantly reduce the disruption to aviation operations. The posterior inversion emission estimates are also sensitive to uncertainty in other eruption source parameters (e.g., the ash density and size distribution) and internal dispersion model parameters (e.g., parameters relating to the turbulence parameterisation). Extending the ensemble inversion methodology to account for uncer-





tainty in these parameters would give a more complete picture of the emission uncertainty, further increasing confidence in these estimates.

## 1  Introduction

Volcanic ash poses a significant risk to aviation as it can cause engines to malfunction, block the system that monitors air speed and external corrosion can reduce visibility (Casadevall, 1994; Clarkson et al., 2016; Clarkson and Simpson, 2017). In the event of a volcanic eruption, authorities need to make fast decisions about which routes are safe to operate and to ensure that airborne aircraft land safely. Safety is of paramount importance but grounding and re-routing of aircraft comes with a large economic cost (e.g., it is estimated the 2010 eruption of the Icelandic volcano Eyjafjallajökull cost the airline industry over £1 billion, Mazzocchi et al. (2010)). Further costs are incurred through the increased maintenance and checks that need to be performed if an aircraft is deemed to have potentially encountered ash. The aim of this paper is to demonstrate the usefulness of applying an inversion technique that optimally combines satellite retrievals and ensemble dispersion simulations to provide an ensemble of the most probable source emission estimates of volcanic ash that will undergo long range transport. Here, the technique is applied to the 2019 Raikoke eruption. This volcano is in a very remote location with no co-located ground based remote sensing that can be used to determine the height of the eruption plume. Also, at the time of the eruption the meteorological situation was rapidly evolving with a large cyclone developing in the North Pacific resulting in the complex filamentation of the volcanic ash cloud. The application of the ensemble inversion approach to this case study demonstrates the benefits of representing the uncertainty in the meteorological data. The results quantify confidence in both the emission estimates and associated ash forecasts.

Currently, the short-range forecast of the geographical location of ash is disseminated to the aviation sector by Volcanic Ash Advisory centres (VAACs) using Volcanic Ash Advisories (VAAs) and Volcanic Ash Graphics (VAGs). These advisories are a combination of output from a Volcanic Ash Transport and Dispersion Model (VATDM), observations of the ash cloud (both from satellites and the ground), pilot reports and forecaster judgement. They indicate the expected location of the ash cloud but contain no quantitative information about ash concentration. Following the 2010 Eyjafjallajökull eruption the UK Met Office (home of the London VAAC) also began producing quantitative peak concentration forecasts for the North Atlantic and European areas. These forecasts use three concentration thresholds which define the following levels of ash contamination: low (200–2000 $\mu$g m$^{-3}$), medium (2000–4000 $\mu$g m$^{-3}$) and high (>4000 $\mu$g m$^{-3}$) (UK Civil Aviation Authority, 2017). These thresholds were determined through consultation between the United Kingdom's Civil Aviation Authority (CAA), Rolls-Royce plc, the UK Met Office, international and European regulators, and aviation experts (Witham et al., 2012; Clarkson et al., 2016). Aviation operators are required to have a safety risk assessment approved by their national aviation authority before aircraft are permitted to fly in regions of medium and high ash contamination, (European Commission, 2011; UK Civil Aviation Authority, 2017). The Roadmap for International Airways Volcano Watch (IAVW) in Support of International Air Navigation (Meteorology Panel International Civil Aviation Organization: Montréal., 2019) states that from 2025 not only will



quantitative ash forecasts need to be provided but also uncertainty information, potentially through the use of ensembles of
VATDM simulations.

The forecasting of ash location and concentration following an eruption is strongly dependent on information about the
eruption that is used to initialise the VATDM simulations (e.g. Webley et al., 2009; Dacre et al., 2011; Stohl et al., 2011;
Tesche et al., 2012; Webster et al., 2012; Harvey et al., 2018; Prata et al., 2019). Typically, VATDMs require the eruption
location, start time and duration, particle size distribution and ash density to be specified. Plus, the time evolution of the height
of the ash plume, the vertical distribution of the ash within the eruption column and the mass eruption rate also need to be
defined. It is possible to estimate the eruption start time using satellites or local observers. There are several remote sensing
techniques to estimate the height of the ash plume (e.g. Oppenheimer, 1998; Petersen et al., 2012). Mass eruption rates are
typically estimated using empirical relationships based on the reported plume height and ash deposits from past eruptions
(e.g. Mastin et al., 2009; Sparks et al., 1997). However, these empirical relationships do not account for other factors that
may influence the plume height, such as the impact of the meteorological situation (e.g., wind bent plumes (Woodhouse et al.,
2013)). This lack of representation of important physical processes and reliance on information from past eruptions can lead to
large uncertainties in the erupted mass estimates. By default, the London VAAC assumes that volcanic ash has a density of 2300
kg $m^{-3}$ and has a particle size distribution based on data from Hobbs et al. (1991). Recent studies by Bruckert et al. (2021) and
Plu et al. (2021) have investigated coupling the detailed plume model FPlume (Folch et al., 2016) to full atmospheric modelling
systems ICON-ART (ICOsahedral Nonhydrosatic - Aerosols and Reactive Trace gases) and MOCAGE (MOdele de Chimie
Atmospherique de Grande Echelle) respectively to determine the impact of a more realistic description of the emissions from
the volcano on the evolution of the simulated ash and sulphur dioxide plume. In both cases, the coupling resulted in a more
realistic representation of the emissions and therefore the horizontal dispersion of the ash plume and a significantly improved
ash forecast.

Meteorological forecast information from numerical weather prediction models are also used as input to VATDMs. This
information includes time evolving 3-dimensional wind fields, precipitation and meteorological cloud location. These meteo-
rological forecasts inform both the transport and dispersion of the ash cloud and the removal of ash from the atmosphere via
wet and dry deposition. The effect of turbulence and small-scale atmospheric motions, which are not resolved in the input
meteorology, are parametrized within the VATDM. Currently, operational VATDMs do not typically consider uncertainties in
the meteorological situation as they only use one realisation of the meteorological forecast. These uncertainties can potentially
lead to significant errors in the forecast ash cloud position and concentration. These errors often occur where ash particles en-
counter regions of large horizontal flow separation in the atmosphere. Ash particle trajectories that originate from very similar
locations can diverge quickly, leading to a reduction in the accuracy of the deterministic forecast (Dacre and Harvey, 2018).
One method of representing this uncertainty is to use an ensemble of meteorological forecasts as input for VATD models. This
approach has been advocated by the volcanic ash community as a way for accounting for wind and precipitation uncertainty
for some time (Bonadonna et al., 2012), however, ensemble meteorology is not routinely used operationally at the VAACs.
This is due to several different barriers including the requirement for VAAs and VAGS to be issued within a prescribed time





window and the need to present the ensemble forecasts in a format which can be used by decision makers to make fast and
robust decisions in an emergency response situation (Mulder et al., 2017).

The use of ensemble meteorology to produce an ensemble of dispersion simulations as a research tool is not new (e.g.
Straume et al., 1998; Galmarini et al., 2004, 2010) but there are only a small number of studies that apply this approach to
volcanic ash forecasts. Recent work by Zidikheri et al. (2018) found that using an ensemble of dispersion model simulations
driven by an ensemble of meteorological fields and different values of ash source parameters gave increased Brier skill scores
at all lead times (the length of time between when the forecast is issued and the time the phenomena are predicted to occur)
compared to a deterministic forecast. A study focussing on the 2013 eruption of Kelut (Dare et al., 2016) found that if an
ensemble is used rather than a single realisation of the meteorological situation there is better qualitative agreement with
satellite observations for lead times greater than 12 hours and similar agreement at short lead times. This is relevant for the
VAAs and VAGs as they are issued out to a forecast lead time of 18 hours. Two earlier studies by Stefanescu et al. (2014) and
Madankan et al. (2014) found that there can be a large spread in predicted ash concentrations at lead times greater than 48
hours when using ensemble meteorology. These studies suggest that the use of ensembles can provide improved volcanic ash
forecasts, however, there is a need to develop strategies to extract information from them that is useful for decision making at
longer lead times.

Satellite imagery in the visible and infrared can show the presence and extent of volcanic ash clouds. Advances in satellite
retrieval techniques mean that estimates of ash cloud top height, effective ash radius and ash column loading are also available
(e.g. Francis et al., 2012; Pavolonis et al., 2013; Grainger et al., 2013). Mass eruption rates at the neutral buoyancy level
can be estimated under certain assumptions (e.g. Woods and Kienle, 1994; Pouget et al., 2013; Prata et al., 2021a) but direct
retrievals of the vertical distribution within the eruption column are not possible. However, satellite retrievals, typically of ash
column loading, can be combined with VATDM simulations using inversion techniques to give time-evolving estimates of these
crucial quantities. There are numerous published approaches that use inversion modelling to estimate ash source parameters for
volcanic eruptions (e.g. Kristiansen et al., 2012; Schmehl et al., 2012; Denlinger et al., 2012; Pelley et al., 2021; Zidikheri et al.,
2017a,b) using a single deterministic realisation of the meteorological situation. This means that uncertainty in the forecast
precipitation or 3d wind fields will lead to uncertainty in the estimated mass eruption rates and their vertical distribution.

As in Harvey et al. (2020), this study brings together inverse modelling and the use of an ensemble of meteorological
forecasts to give an ensemble of the most probable source emission estimates of volcanic ash that will undergo long range
transport following the 2019 Raikoke eruption. These emission estimates can be used to obtain robust ash forecasts constrained
by observations. There will be a particular focus on regions where medium and high levels of ash contamination are predicted,
as these are areas that aircraft may be prohibited from entering, and on the influence of emissions of ash into the stratosphere
on these regions.

The methods and data used in this study are described in Section 2. Section 3 describes the details of the 2019 Raikoke
eruption. The volcanic emission estimates determined using the ensemble inversion system, their impact on volcanic ash
forecasts and on flight planning decisions are presented in Sections 4 and 5. A summary, conclusions and implications for
future work are presented in Section 6.





## 2 Methods and data

### 2.1 Ensemble of Meteorological Forecasts

The Met Office Global and Regional Ensemble Prediction System (MOGREPS-G) has 17 ensemble members plus a control member. It has a horizontal resolution of 20 km in mid-latitudes and there are 70 vertical levels with the lid at approximately 80 km. Each forecast is initialised 4 times per day at 0000, 0600, 1200 and 1800 UTC and they extend out for 7 days (Bowler et al., 2008). At the time of the Raikoke eruption, MOGREPS-G used a stochastic physics scheme to account for model uncertainty and an online inflation factor calculation to calibrate the spread of the ensemble in space and time (Flowerdew and Bowler, 2011, 2013). The MOGREPS-G forecasts used to drive NAME in this study are initialized at 1200 UTC 21 June 2019.

### 2.2 Satellite observations

#### 2.2.1 Himawari-8

Himawari-8 is a geostationary satellite with 16 spectral channels that came into operation in July 2015 (Bessho et al., 2016). Its high temporal (10 min) and spatial (2 km at nadir for the infra-red bands) resolution makes its observations ideally suited to evaluate the transport of volcanic ash following an eruption. Two independent volcanic ash retrieval algorithms, one based on work primarily from the UK Met Office (Francis et al., 2012) and one determined using the methodology described in (McGarragh et al., 2018), are used in this study and are described below. Note that although the retrieval methods have been developed independently in different research groups, they both use information from the Himawari-8 satellite and therefore have the same bias when volcanic ash is obscured by meteorological cloud (which is prevalent during the Raikoke 2019 eruption).

**Met Office Algorithm**: This algorithm is based on the method described in Francis et al. (2012), with slight adaptations for the channels of the Advanced Himawari Imager (AHI). Firstly, the channels at 8.6 $\mu$m, 10.4 $\mu$m and 12.4 $\mu$m are used to detect which pixels are contaminated by volcanic ash. Then the ash column loading, layer height and effective radius are determined using a one-dimensional variational (1D-Var) analysis to determine an optimal estimate between the assumed background and the observed radiances in the channels at 10.4 $\mu$m, 12.4 $\mu$m, and 13.3 $\mu$m. The detection is based on a combination of brightness temperature difference (BTD) tests and beta ratio tests that are optimised for the June 2019 Raikoke eruption. The beta ratio tests use a derived radiative parameter $\beta$, originally devised by Pavolonis (2010), that is the effective absorption optical depth ratio of two channels and are used to filter pixels marked as ash by the BTD tests. To reduce false detections over arid land surfaces and at high satellite zenith angle, several geographical filters are used. Checking the consistency of ash detection in neighbouring pixels also removes other false detections.

Where ash is detected, the Met Office algorithm determines the ash column loading. These pixels are flagged as containing ash. If a pixel is free from both ash and meteorological cloud then it is flagged as a clear sky pixel. Pixels that don't have detectable ash and are not flagged as clear skies are unclassified. As in Pelley et al. (2021), further processing is performed to regrid the retrieved column loadings on to a grid of 0.375° latitude by 0.5625° longitude (approximately 40 km × 40 km in





mid-latitudes) and averaged over 1 h. This is to match the resolution of the NAME ash column loading output and to reduce data volumes. If 50% or more satellite pixels in a grid box contain ash or more than 90% of pixels are classified as ash or clear skies, then the grid box is selected for use in the InTEM inversion. If all classified pixels within a grid box are flagged as clear sky pixels, then the grid box is deemed to be a clear sky observation. Otherwise, the grid box is deemed to be an ash grid
observation with the column loading in this grid box given by the mean of all the classified pixels (including clear skies).

**ORAC Algorithm**: To estimate the mass loading of fine ash for the Raikoke ash clouds, the Optimal Retrieval of Aerosol and Cloud (ORAC, McGarragh et al., 2018) algorithm was applied to infrared measurements made by the AHI on board the Himawari-8 satellite. ORAC uses the optimal estimation approach (Rodgers, 2000) to retrieve state variables based on radiative transfer simulations, satellite measurements and prior information. The ORAC algorithm includes optical depth,
effective radius, surface temperature and cloud-top pressure in the state vector and the mass loading is derived from the retrieved optical depth and effective radius consistent with previous authors (e.g. Wen and Rose, 1994; Corradini et al., 2008; Prata and Prata, 2012). To retrieve volcanic ash properties consistently over day and night the 10.4, 11.2, 12.4 and 13.3 $\mu$m thermal infrared channels are used in the measurement vector. The microphysical model used in the radiative transfer model assumes an ash cloud containing spherical particles which conform to a log-normal distribution and ash composition
based on the Eyjafjallajökull ash taken from the Aerosol Refractive Index Archive (ARIA, http://eodg.atm.ox.ac.uk/ARIA/). Ash detection follows the approach presented in Appendix A of Prata et al. (2021b) and only retrievals that converge with a measurement cost at solution of less than 10 are considered. Four forward model configurations are used in the retrievals: 1) A single tropospheric layer of ash, 2) A tropospheric ash layer above a liquid water cloud layer, 3) A single stratospheric layer of ash and 4) A stratospheric ash layer above a liquid water cloud layer. The multi-layer configurations are introduced to account
for prevalent low-level stratus clouds during the explosive phase of the Raikoke eruption. The water cloud layer is tightly constrained with an *a priori* cloud-top pressure of 800hPa, effective radius of 10 $\mu$m and 550 nm optical depth of 16. The *a priori* uncertainties on these values are set to 50 hPa, 1 $\mu$m and 3. The *a priori* settings were chosen based on ORAC standard cloud retrievals (Poulsen et al., 2012; McGarragh et al., 2018) that are run separately on the nearby, low-level stratus clouds. The troposphere/stratosphere model configurations are run by setting two different a priori cloud-top pressures to determine
cloud-top heights above and below the tropopause.

The *a priori* cloud-top pressures considered are 500 hPa and 200 hPa. The stratospheric *a priori* cloud-top pressure is chosen based on independent CALIPSO observations of the stratospheric ash cloud. After running all four forward model configurations, we select the retrieval with the lowest cost for each ash affected AHI pixel to generate the final retrieval product. Further details of the algorithm used to generate the retrievals shown here are given in Prata et al. (In preparation,
185   2021).

## 2.3   NAME

In this study the Numerical Atmospheric-dispersion Modelling Environment (NAME) model was used to simulate the dispersion of volcanic ash (Jones et al., 2007). To model the transport and removal of volcanic ash, NAME includes parameterizations of dispersion due to free tropospheric turbulence, sedimentation, dry deposition and wet deposition. It is assumed that the ash





particles are spherical and have a density of 2300 kg m$^{-3}$, although in reality the density of the ash is determined by its porosity, chemical composition and grain size. In this study, aggregation of ash particles, near source plume rise and processes driven by the eruption dynamics (e.g., Woodhouse et al., 2013) are not explicitly modelled. The particle size distribution used is based on data from Hobbs et al. (1991), but only includes ash particles with diameters between 1 - 30 $\mu$m, as ash particles larger than this are not typically detected by the AHI. We refer to this as *fine ash* in this paper.

## 195    2.4    InTEM for volcanic ash

The Inversion Technique for Emissions Modelling (InTEM) for volcanic ash is an inversion system that combines VATDM simulations, satellite retrievals and a prior estimate of the emission using a Bayesian approach to give the best estimate of the emissions profile for fine ash that can undergo long range dispersion. This system has been developed at the UK Met Office and was originally developed to estimate greenhouse gas emissions (Manning et al., 2011). It has been previously used by Harvey

et al. (2020) to assess the impact of ensemble meteorology on estimates of volcanic ash emissions from the 2011 Grímsvötn eruption. The posterior emission profile can either be determined using satellite retrievals of ash only or of both ash and clear skies and has a chosen vertical resolution of 4 km and a time resolution of 3 hours. Full details of the InTEM system for volcanic ash are given in Pelley et al. (2021) and Harvey et al. (2020).

### 2.4.1    Estimate of the prior source term

To ensure that the inverted source term is not overfitted to the satellite information, a prior estimate of the source term is used. This also makes sure that the posterior source term is informed by known information about the eruption, such as the eruption start and end time and the maximum plume height. To construct the prior, it is assumed that emissions in the eruption column are uniform in the vertical from the volcano vent to an estimate of the plume height based on observations with an error of +/- 2km. The mass eruption rate is determined using the empirical relationship in Mastin et al. (2009). The mean and error

covariance matrix of the prior are estimated using a stochastic model that includes correlations between errors in the emissions at different heights and times. Thomson et al. (2017) contains a full description of how the prior is determined and used in the InTEM system. In this study the prior source term is based on a plume height of 13km, which is consistent with information provided by the Tokyo VAAC and Bruckert et al. (2021). Note that the prior is a probability distribution and from now on it is assumed that when the prior is mentioned refers to the mean of this distribution.

### 215    2.4.2    VATDM simulations

The VATDM simulations within InTEM are performed using NAME. As this methodology exploits ash column loadings determined from satellite retrievals, the NAME simulations only include ash particles with diameters between 1 - 30 $\mu$m as ash particles larger than this are not typically detected by the AHI. This assumption is consistent to those made in other inversion systems for this application (e.g., Stohl et al., 2011). Simulations representing a nominal release rate (1 g $s^{-1}$) from each

possible source term component (4km height range and 3-hourly time period) are conducted. Model predictions of ash column





loads can be easily determined for an arbitrary emission profile by a linear combination of these nominal simulations. The resolution of the posterior emission estimate (3-hourly with 4km vertical resolution), is chosen to ensure that the inversion can be performed within a time frame that is compatible with VAAC operational constraints.

### 2.4.3   The inversion algorithm

The prior estimate, NAME simulations and satellite retrievals are combined within the inversion scheme to give a posterior distribution of emissions. The posterior distribution is Gaussian and within InTEM the best estimate of emissions is taken as the peak of this distribution with a non-negative constraint applied. It is possible that selecting the peak of the distribution ignores a large amount of uncertainty information within the posterior and results in a loss of information about the emissions, but this is not the focus of the present study. A quadratic cost function, representing the simultaneous fit of the VATDM simulations

and satellite retrievals, and between the emission estimate and the prior, is minimised using the Lawson and Hanson (1974) non-negative least squares algorithm. This algorithm was chosen as it converges in a finite number of iterations and is very fast. In this study the inversion algorithm took less than 1 minute to run for each of the ensemble members using satellite retrievals from 1800 UTC on 21 June to 0000 UTC on 22 June 2019. The speed of the InTEM system is therefore governed by the length of time it takes to perform the VATDM simulations. The best estimate of the emissions determined by InTEM can then

be used as the ash emission profile in simulations used to forecast the evolution of the volcanic ash cloud. From now on, it is assumed that when posterior emissions are used it is the peak value. Pelley et al. (2021) and Thomson et al. (2017) contain a full description of the inversion scheme.

## 3   Raikoke 2019: Case study description

Raikoke is an uninhabited volcanic island near the centre of the Kuril Island chain in the Sea of Okhotsk in the northwest Pacific

Ocean at 48.2°N, 153.3°E. The volcano has a vent height of 551 m. Its most recent explosive eruption started at 1800 UTC on 21 June 2019 when a series of nine explosive events occurred until approximately 0600 UTC on 22 June. It is estimated that the initial eruptive plume height of 10-14 km asl (Global Volcanism Program, 2019a) and there is evidence in visible satellite imagery of an umbrella cloud. Ash and sulphur dioxide were dispersed by the jet stream as well as being entrained by a cyclone located near the Komandorskiye Islands. Forty aeroplanes were diverted because of the ash plume produced by this eruption

(Global Volcanism Program, 2019b).

## 4   Results

### 4.1   Posterior inversion estimates

Figure 1 shows the posterior height-time ash emission rates for particles in the range between 1 - 30 $\mu$m obtained using InTEM for the Raikoke eruption (21-22 June 2019) using each of the MOGREPS-G meteorological ensemble members with Himawari

retrievals of ash and clear skies from 21 - 24 June. Each panel shows the ash emission rates determined by InTEM using a single





member of the MOGREPS-G ensemble. The vertical emission profiles are similar but there are differences in the magnitude of ash emitted. Four members (e.g., member 13) have continuous emission of ash between 4-12 km above vent level (avl) whereas the other 14 members have times when there is no emission of ash at this height range. There is very little ash emitted between 0 and 4 km avl in all members which is qualitatively consistent with the visible satellite imagery indicating an umbrella cloud

shape (although it is important to note that the NAME simulations presented here do not represent the radial spreading of an umbrella cloud which is a source of uncertainty in the determination of the posterior emissions). There is a range in the total emissions of fine ash over the entire eruption of 0.32 – 0.71 Tg (shown in Figure 2 as blue circles), with an ensemble mean value of 0.52 Tg. We can compare the ensemble mean value with other estimates of total fine ash estimated from measurements from the Advanced Himawari Imager. Muser et al. (2020) estimate values of 0.4 – 1.8 Tg and Capponi et al. (2021) estimate

an ensemble mean total fine ash of 0.49 Tg. Both of these estimates are quantitatively similar to the mean value of total fine ash estimated from the InTEM ensemble presented here. Although, it is important to note that a deck of stratus cloud was present at low levels (approximately 2km) potentially leading to an underestimation of the amount of ash present in the lowest part of the atmosphere. There is also a range, 4459–5314, in the number of unique observations which impact the inversion between ensemble members. This represents the variability between the ensemble meteorology that is used in the VATD model

simulations used in the inversion process.

Figure 3 (a) shows the prior emission profile which was determined using the empirical Mastin relationship (Mastin et al., 2009) and a plume height of 13 km avl which is consistent with reports from the Tokyo VAAC and the plume heights used in Bruckert et al. (2021). The emission rates are greater at all heights and times when compared to the posterior estimates. The prior total emission is 11 Tg, which is approximately 15.5 times larger than any of the posterior estimates (shown in Figure 2).

Figure 3 (b) shows the ensemble mean posterior emission profile which shows this large reduction in emissions at all levels especially below 4 km avl where ash is only emitted for one 3 hour period at the start of the eruption. This reduction greatly impacts the fraction of emissions released near the surface (0-4 km), in the troposphere (4-12 km) and in the stratosphere (above 12 km) as seen in Figure 4. (Note the heights chosen here to define the vertical levels in the atmosphere are based on the vertical resolution of the InTEM emission profile and are not standard meteorological definitions.) The prior emission profile

has 31% of the ash emitted near the surface, 61% in the troposphere and 8% in the stratosphere compared to the posterior ensemble mean of 3% of the ash emitted near the surface, 63% in the troposphere and 34% in the stratosphere. Fractionally there is a much larger amount of ash emitted into the lower stratosphere in the posterior ensemble. This has consequences for the temporal evolution of the simulated ash cloud as ash in the stratosphere cannot be easily deposited to the surface through wet deposition and sedimentation. Plus, due to vertical wind shear, ash within the stratosphere can be transported by

atmospheric winds with different speeds and directions compared to other parts of the atmosphere. Simulations of the evolution of the Northern Hemisphere mean sulphur dioxide mass burden performed by de Leeuw et al. (2020) were also found to be sensitive to the amount of sulphur dioxide emitted in the stratosphere. The NAME simulation with the best agreement with satellite retrievals of sulphur dioxide from TROPOMI in terms of peak concentrations and e-folding times had a larger fraction of sulphur dioxide emitted in the lower stratosphere compared to the control simulation.


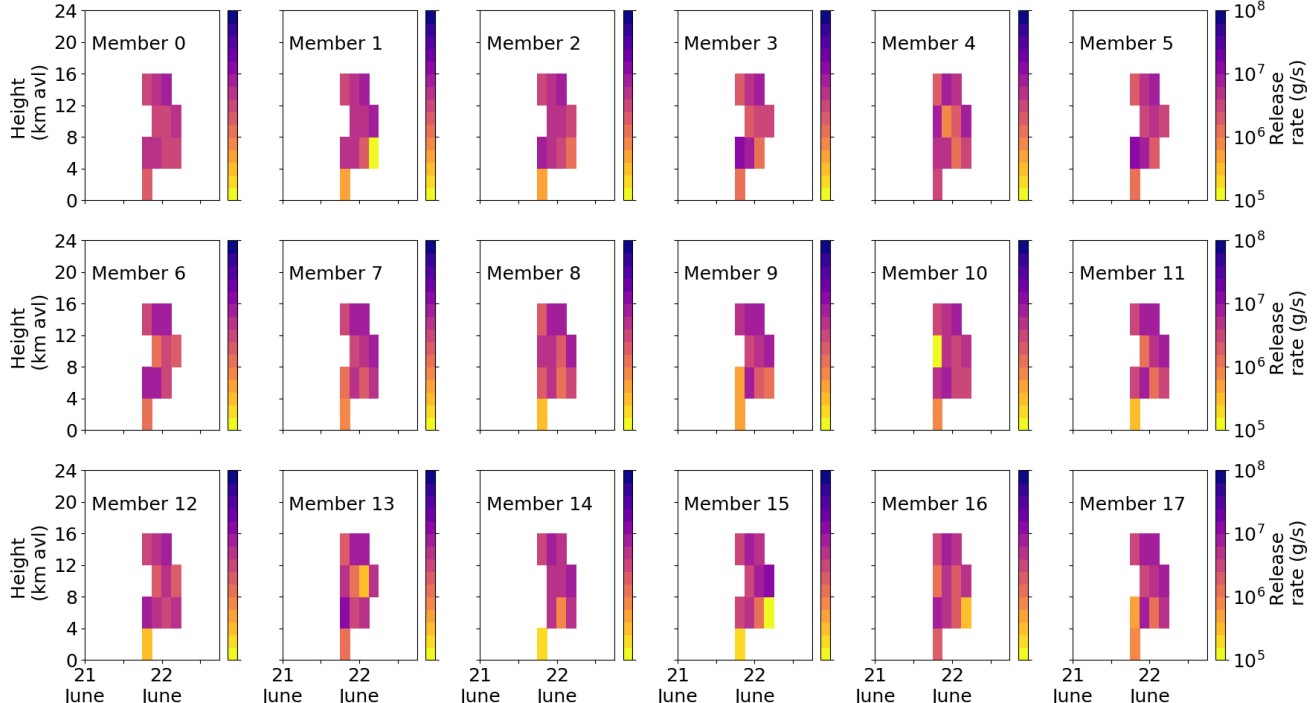

**Figure 1.** Posterior emission profiles (g/s) estimated by InTEM for the 2019 Raikoke eruption for each member of the MOGREPS-G ensemble using Himawari retrievals of ash and clear skies. Note the logarithmic colour scale.

## 285  4.2  Ensemble InTEM ash forecasts

The previous discussion in Section 4.1 focused on the differences in posterior emission profiles, however VATDM forecasts of the ash cloud are used to inform VAAC graphics and advisories. This section analyses NAME simulations driven with the MOGREPS-G meteorological ensemble both using the peak of the posterior distribution of emissions (shown in Figure 1) and prior emissions and evaluates these simulations against independent satellite retrievals using the ORAC Himawari retrievals.

This analysis is performed to determine if the observation constrained posterior emissions estimate results in a more accurate ash forecast than when using the prior emissions when evaluated against an independent set of satellite retrievals. Note that the posterior emissions are determined using satellite retrievals for the period 21-25 June 2019 and not just those retrievals available before 1200 UTC 22 June 2019 (Figure 5) and 1200 UTC 23 June 2019 (Figure 6).

Figures 5a-b show the spatial extent of the ash cloud at 1200 UTC 22 June 2019 for the ensemble of VATDM simulations
with posterior emissions determined using satellite retrievals obtained between 21 and 25 June 2019 (Figure 5a) and prior emissions (Figure 5b). Figures 5c-d show Himawari retrievals using the Met Office retrieval algorithm (Figure 5c) and the ORAC retrieval algorithm (Figure 5d). There is a large difference between the magnitude of the ash column loadings in the simulated ash clouds in Figure 5a and b. The mean ash cloud extent from the prior ensemble is similar to the posterior ensemble



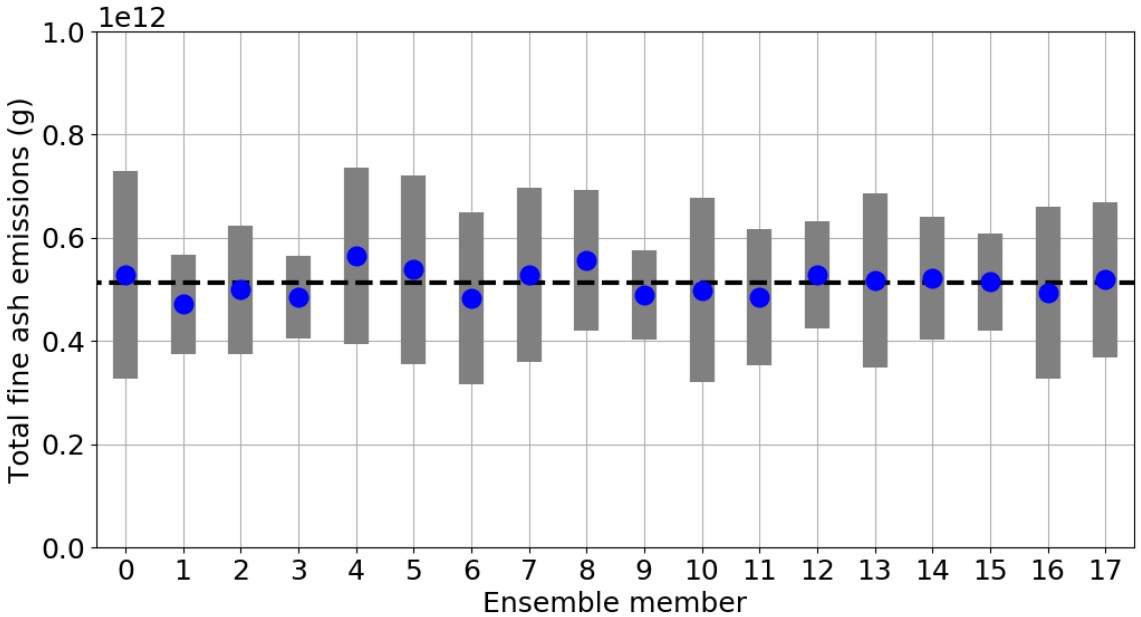

**Figure 2.** Total fine ash emitted during the 2019 Raikoke eruption for each member in the MOGREPS-G ensemble determined using InTEM. Blue circles and grey bars indicate the peak in the posterior distribution and range ($\pm$ one standard deviation) of the total ash emitted. The black dashed line indicates the ensemble mean total fine ash emissions.

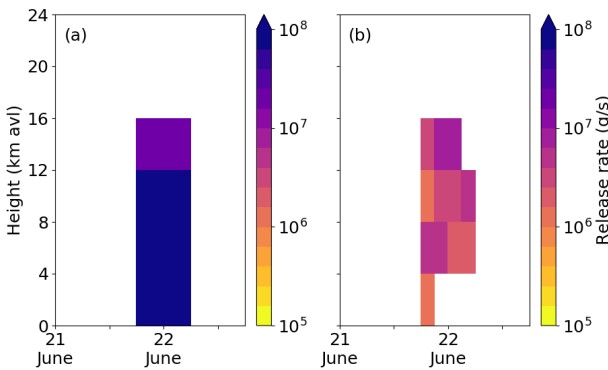

**Figure 3.** (a) Prior (b) ensemble mean posterior time-height emission profile (g/s) determined by InTEM for the Raikoke eruption. Note the log scale used for the release rate.

but with column loading magnitudes that are more than 10 times larger. In both ensembles, the highest column loadings are in
similar locations within the ash plume with ensemble mean values approximately 68 g m$^{-2}$ in the prior ensemble and 4.8 g
m$^{-2}$ in the posterior ensemble. By design the posterior ensemble mean column loading values are quantitatively closer to the





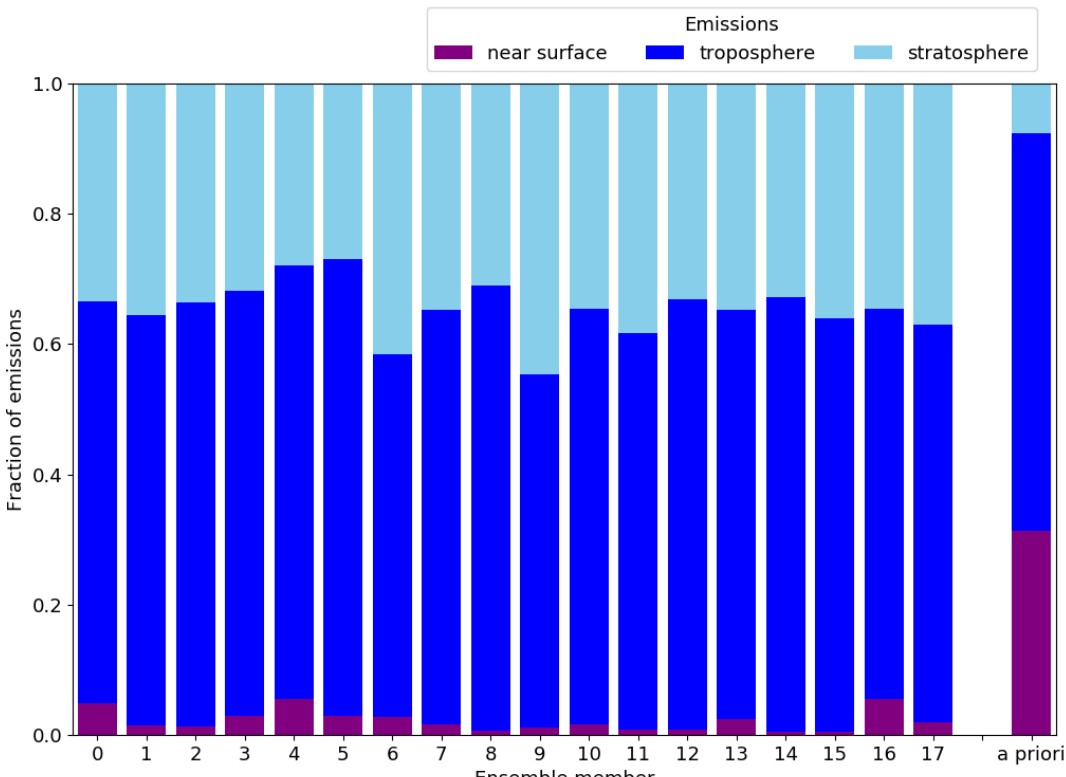

**Figure 4.** The fraction of total ash during the 2019 Raikoke eruption emitted near the surface (0-4km), in the troposphere (4-12km) and in the stratosphere (above 12km) for each member in the MOGREPS-G ensemble determined using InTEM. The final bar shows the fraction of ash emitted near the surface, tropopause and stratosphere in the prior emission profile.

satellite retrievals used in the InTEM system (Figure 5c) but they are also closer to the independent ORAC retrievals (Figure 5d) than the prior ensemble mean column loading values.

Figure 6a-b show the difference between the simulated ash clouds at 1200 UTC on 23 June 2019 with InTEM derived
posterior estimates using satellite retrievals until 0000 UTC 25 June 2019. This is 24 hours later than the ash cloud shown in Figure 5. At this time the ash cloud is much more extensive and both the prior and posterior ensemble mean plume extends as far as 164° W. The ash plume is also entrained into a cyclone that was present in the North Pacific at this time. This is most evident in the prior ensemble where the ensemble mean column loadings are approximately an order of magnitude higher than the posterior ensemble mean column loadings.

Producing an ensemble of VATD simulations allows the assessment of the ensemble spread. Figure 5e shows the range of posterior column loading values for the 18 posterior emission simulations (blue line and shading) and 18 prior ensemble simulations (cyan line and shading) along the cross section shown in panels (a-d) as a black dashed line. The coloured shading



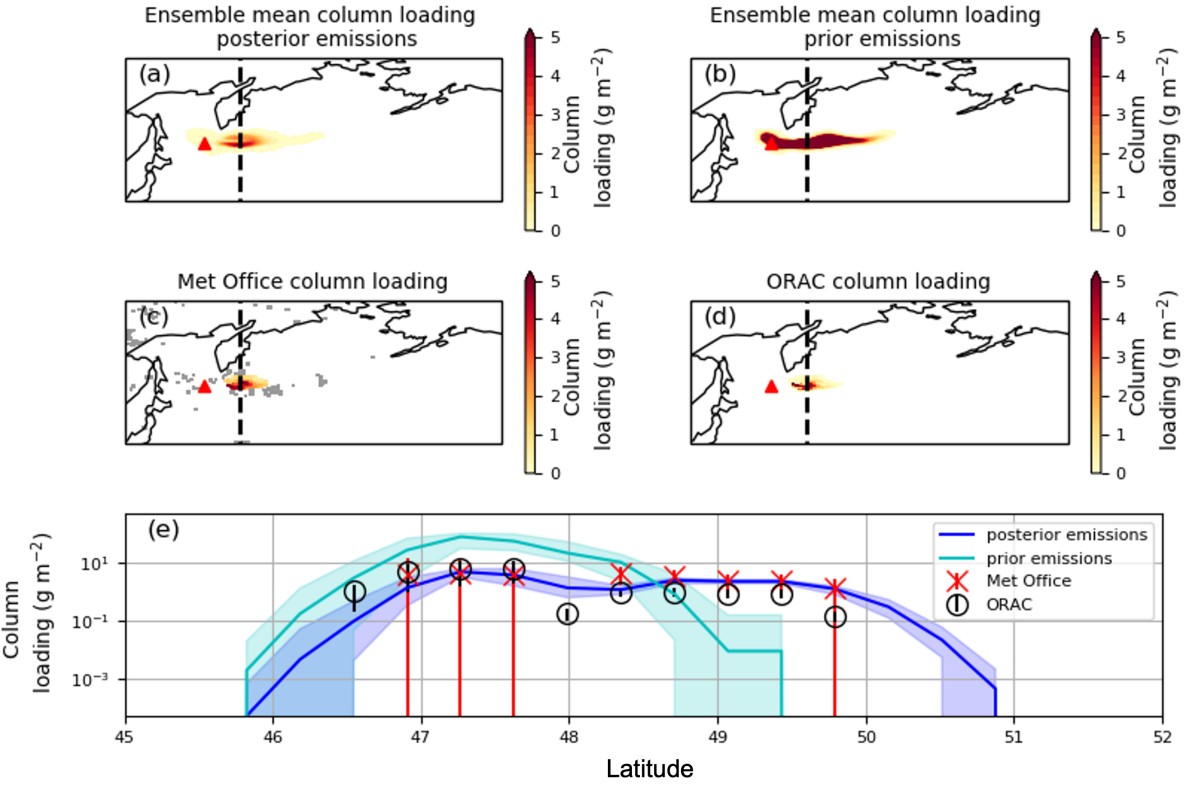

**Figure 5.** (a) Ensemble mean ash column loading for simulations run with emission profiles determined using InTEM with matching ensemble meteorology at 1200 UTC 22 June 2019, (b) Ensemble mean ash column loading at 1200 UTC 22 June 2019 for simulations run with prior emission profile and ensemble meteorology, (c) Himawari ash column loading retrieved using the Met Office algorithm (grey shading indicates grid boxes that are classified as clear sky) (d) Himawari ash column loading retrieved using the ORAC algorithm for 1200 UTC 22 June 2019, (e) Ash column loading profile from 45 - 52 °N along the cross section indicated by the black dashed line in panels (a)-(d). Posterior ensemble (blue), prior ensemble (cyan), Himawari ash column loading retrieved using the Met Office algorithm (red crosses) and Himawari ash column loading retrieved using the ORAC algorithm (black circles). The error bars represent ± one standard deviation in the estimates of ash column loading determined from the satellite retrievals. The coloured shading indicates the range of column loading in the ensemble.

represents the uncertainty in ash column loading due to the meteorological ensemble. Satellite retrievals of ash column loading and associated uncertainty from Himawari using the Met Office algorithm (red crosses) and from Himawari using the ORAC algorithm (black circles) are shown for comparison. The Met Office and ORAC retrievals, at this time and along this cross section, are very similar. Between 45-50° N, the posterior ensemble spread encompasses the Met Office retrievals within their retrieval uncertainty. Between 46.5-48.5° N, the posterior ensemble spread falls within the uncertainty of the ORAC retrievals.




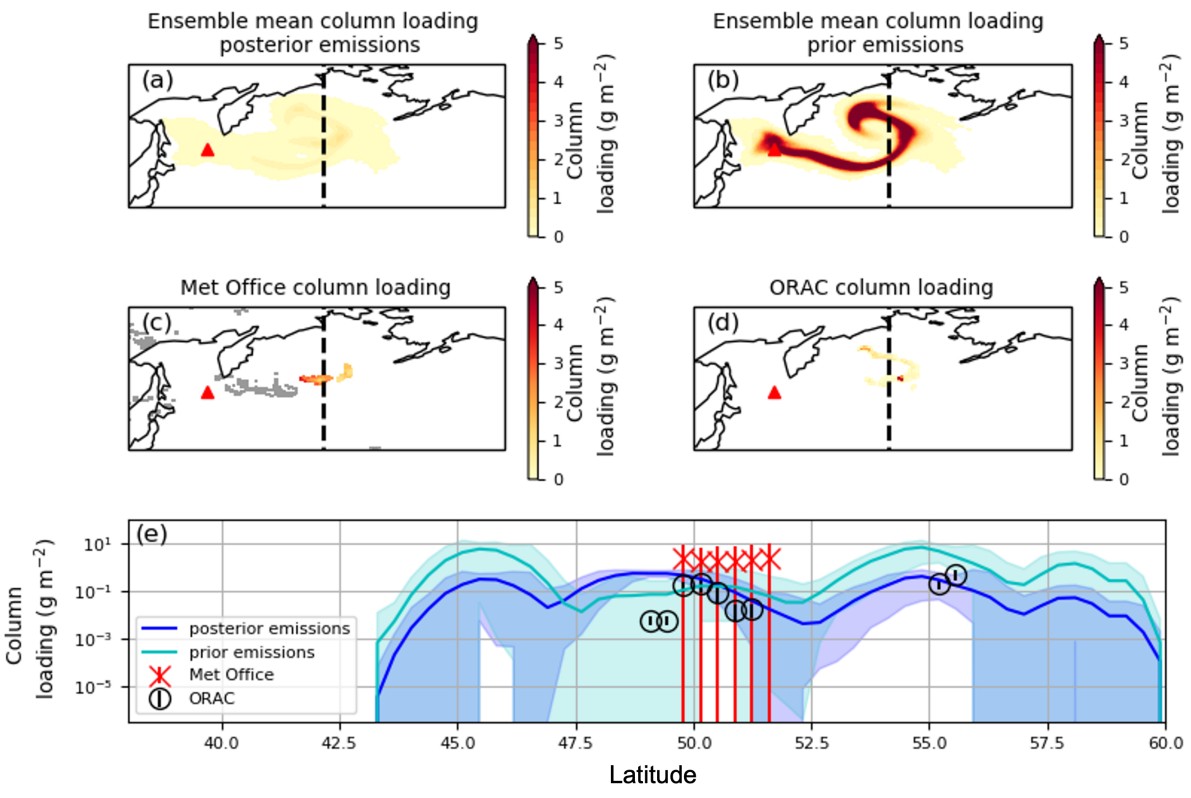

**Figure 6.** As Figure 5 for 1200 UTC on 23 June 2019.

The prior ensemble has a much higher mean magnitude than the posterior emissions (by over a factor of 10) and the ensemble spread does not encompass either set of satellite retrievals within one standard deviation. Meteorological cloud obscures much of the domain so obtaining a contiguous retrieval of ash and clear sky pixels is not possible at this time.

Figure 6e shows the same variables as Figure 5e at 1200 UTC 23 June 2019. The magnitude of the column loading in the VATD simulations is approximately 10% of the column loadings shown in Figure 5. The prior ensemble column loadings have a large spread in forecast ash column loadings and are indicated by the coloured shading. This spread shows the large variability that can be introduced by using an ensemble of meteorological conditions with the same ash emission profile. In this case the cross section intersects the simulated ash plume in 3 locations - 45-46°N, 49-51°N and 53-59°N. However, the Himawari retrievals used in the inversion only detects ash 47-52°N. At this location, both the prior and posterior column loadings lie within the uncertainty of the ash retrievals used in the inversion with the prior emissions lying within one standard deviation of the independent ORAC Himawari ash retrievals. At 55-56°N, the posterior ensemble column loadings are within the uncertainty of the ORAC retrievals but no ash is detected using the Met Office algorithm.





### 4.3 Vertical distribution of emissions

To assess the impact of a larger fraction of ash being emitted into the stratosphere we focus on the evolution of the ash plume using inverted emissions and matching meteorology for ensemble members 5 and 9. These members were chosen as member 5 has the smallest fraction of ash emitted into the stratosphere (27%), whereas member 9 has the largest (44%). To determine the differences between the simulations driven by emissions from members 5 and 9, the output from the NAME simulations is post-processed by dividing the vertical grid into three layers (near surface, troposphere and stratosphere) and setting the concentration of each of the three layers to the maximum ash concentration of the original higher resolution layers. Figure 7 shows the maximum over the simulated ash layer concentrations near the surface, in the troposphere and stratosphere at 1200 UTC on 23 June 2019. At this time, the ash plume structure at all three levels is qualitatively very similar. However, there are some differences both in plume location, extent and ash concentration values. The location differences are likely to be due to the use of different driving meteorology. Near the surface and in the troposphere, member 5 has higher maximum concentrations than member 9 but in the stratosphere the opposite is true. This is expected as more mass is emitted at these heights in member 9. The differences in peak concentration are largest when considering the stratosphere (2.53 g m$^{-3}$ in member 5 compared to 3.32 g m$^{-3}$ in member 9). In the stratosphere member 9 has the highest peak concentration despite the plume extending further east (by approximately 8 degrees) and the plume having an area that is 1.4 times larger than the plume produced using member 5 and the region with concentrations greater than 200 $\mu$g m$^{-3}$ being 1.5 times larger than the equivalent region in member 5. These differences highlight the importance of the vertical distribution of the ash emissions as it can lead to an increase in areas that may be considered too contaminated for aircraft to fly through. Note that the lowest ash concentration shown in Figure 7 is 100 $\mu$g m$^{-3}$.

### 5  Potential implications for aviation operations

The comparison between simulated and satellite retrievals of ash column loadings is valuable for forecast validation, however this variable does not give any information about the vertical extent of the ash cloud or peak ash concentrations at the three International Civil Aviation Organisation (ICAO) prescribed flight levels. Numerous charts are required to visualise the ash extent at different flight levels, different concentration levels and different times. During an emergency, the number of graphics associated with an ensemble of simulations can be overwhelming, plus the interpretation of ensemble spread relies on the decision makers experience and risk appetite (Mulder et al., 2017).

One approach of condensing this data into a single chart using a risk matrix is presented by Prata et al. (2019). Here we apply the same risk-based approach to the Pacific region following the eruption of Raikoke using both the prior and posterior ensembles outlined in Section 4. The use of this type of graphic reduces state-of-the-art ensemble information into an easy to interpret decision making tool that can be used to make fast and scientifically robust decisions. As in Prata et al. (2019), geographical regions that are considered potentially hazardous to aircraft are identified based on the fraction of ensemble members that exceed low, medium and high ash concentrations. The concentration thresholds used are defined by the UK CAA (UK Civil Aviation Authority, 2017) and are 200-2000 $\mu$g m$^{-3}$ (low), 2000-4000 $\mu$g m$^{-3}$ (medium) and >4000 $\mu$g m$^{-3}$

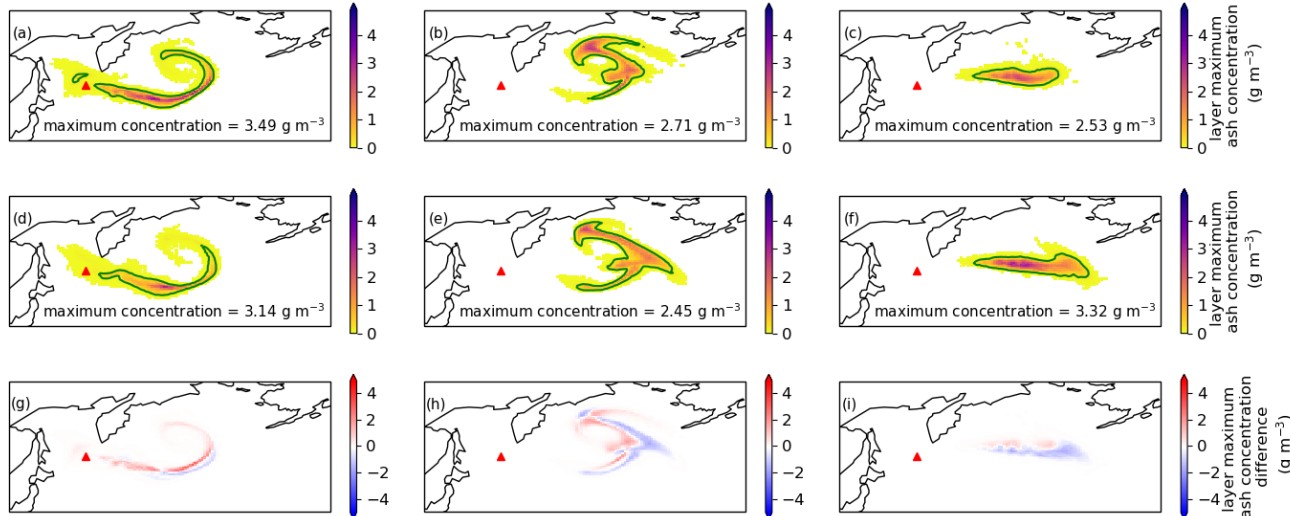

**Figure 7.** Maximum layer ash concentration (g m$^{-3}$) for (a,d) near surface (0-4 km), (b,e) troposphere (4-12 km) and (c,f) stratosphere (above 12 km) for ensemble member 5 (a-c) and member 9 (d-f) at 1200 UTC on 23 June 2019. The green contour denotes the 200 $\mu$g m$^{-3}$ concentration level. Panels (g-i) show the difference between maximum layer concentrations for member 5 and member 9 also at 1200 UTC on 23 June 2019. The lowest ash concentration shown in (a-f) is 100 $\mu$g m$^{-3}$

(high). This is done for each of the three VAAC flight levels and the overall risk at a given location is the maximum risk over the three flight levels. This is a conservative approach but is in line with the current ICAO guidance (International Civil Aviation

Organization: Montréal., 2007). In this analysis, to be consistent with the approach in Prata et al. (2019), the ash concentration fields output by NAME were multiplied by a factor of 10, known as the "peak-to-mean" factor. This factor accounts for peak concentrations that are not resolved in the NAME simulations.

Figure 8 shows the risk determined using the Prata et al. (2019) approach using both the posterior emissions ensemble and the prior ensemble at 1200 UTC 22 June 2019 and 1200 UTC 23 June 2019. These ensembles represent the uncertainty in ash

location and ash concentration due to uncertainty in the meteorological situation. At both times, the region of forecasted risk is reduced when the posterior emission ensemble is used compared to the prior ensemble. At 1200 UTC 22 June the forecasted risk area is reduced by 31%, with the highest risk area (blue) reduced by a similar amount when the posterior ensemble is used. At 1200 UTC 23 June, the impact of the cyclone in the Pacific can still be clearly seen in the risk associated with both ensemble members. At this time, the forecasted risk area is reduced by 35%, with the highest risk area reduced by 49%. It

is also possible to see the potential impact on a hypothetical flight track between San Franciso (SFO) and Shanghai (PVG) International airports (shown as a grey lines on each panel). At 1200 UTC 22 June the track would not directly transit regions of ash risk as the plume at that time has a relatively small extent. By 1200 UTC 23 June, the flight track would encounter the ash plume when both the prior and posterior ensembles are used, with the percentage of the route from PVG to SFO impacted





by the highest risk reduced from 11% to 7% when the posterior ensemble is used instead of the prior. For med-level risk, the
percentage is reduced from 17% to 11%. At both times, the application of this risk-matrix approach highlights the potential
impact of using the ensemble inversion approach on airline operations. Disruption could have been reduced and the high
economic cost actions (such as flight cancellation, rerouting and enhanced engine checks) could have been greatly decreased.

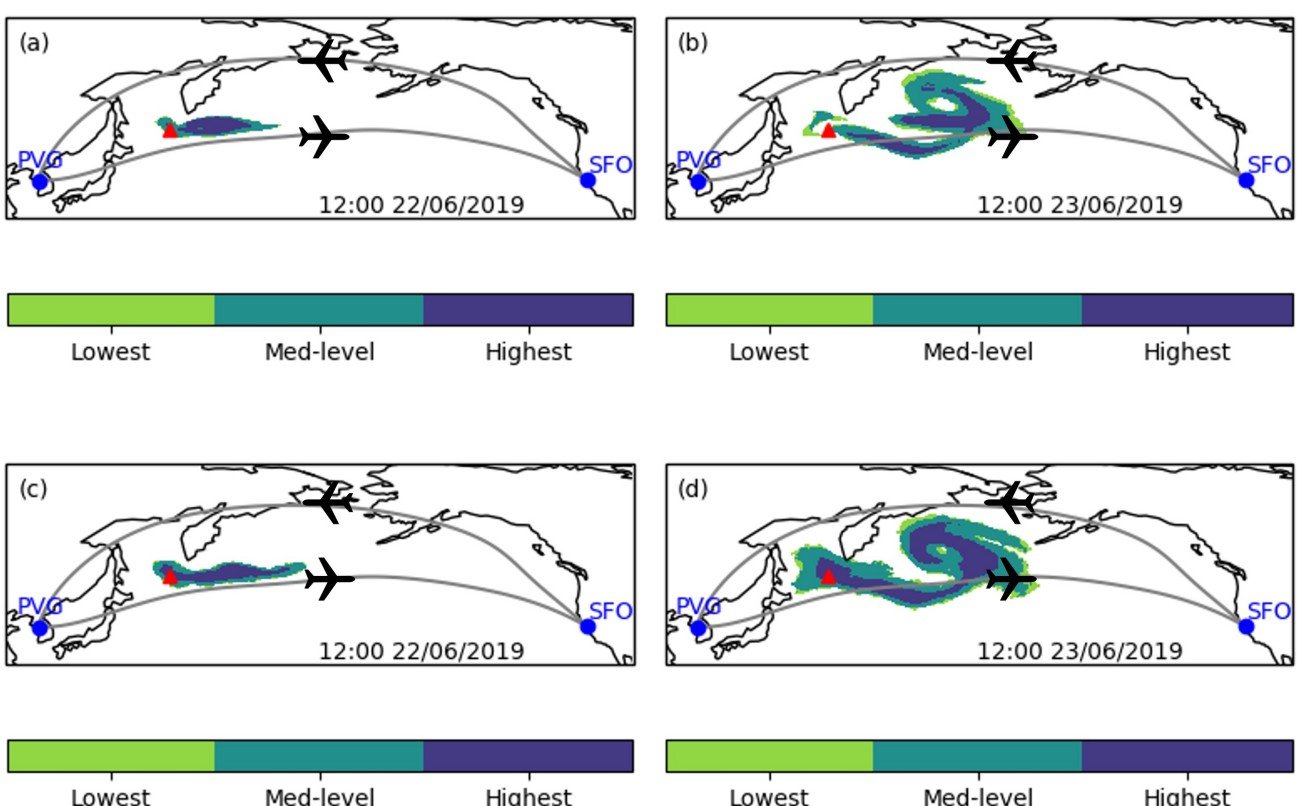

**Figure 8.** Overall ash concentration risk map at (a,c) 1200 UTC on 22 June 2019, (b,d) 1200 UTC on 23 June 2019 for (a,b) simulations
with posterior emissions with matching ensemble meteorology and (c,d) simulations with prior emissions with ensemble meteorology. Green
shading indicates the lowest level of risk, turquoise shading indicates med-level risk and purple indicates the highest level of risk. The grey
line indicates a hypothetical flight route between San Franciso (SFO) and Shanghai (PVG) International airports with the black aeroplane
icons indicate the direction of travel between them.




## 6 Conclusions

The eruption of Raikoke on 21 June 2019 sent volcanic ash high into the atmosphere. In this study satellite retrievals from
the Himawari satellite and an ensemble of NAME simulations driven by an ensemble of meteorological forecasts have been
combined using the InTEM inversion system. Our main results can be summarized as follows:

- For this case study, the posterior ash emission rates determined using InTEM are substantially lower compared to the
  prior emission profile estimated using the Mastin et al. (2009) relationship. The posterior emission profiles produced
  using a range of plausible meteorological situations are qualitatively very similar giving confidence in the use of the
  InTEM system. However, there are differences in the magnitude of the ash emitted at different heights. There is a large
  range in the fraction of mass that is emitted into the stratosphere (above 12 km avl in this study). These differences
  lead to a range of values (0.32 – 0.71 Tg) for the total amount of ash (in the size range 0.1–100 $\mu$m) emitted over the
  eruption period. This range is broadly consistent to the range found in Muser et al. (2020). It should be noted that the
  reduction in emissions determined by InTEM compared to using the Mastin et al. (2009) approach is much larger than
  the differences between the emissions determined with the ensemble of meteorological situation. In this case, this points
  to the Mastin et al. (2009) relationship giving ash emissions that are grossly over estimated. However, the Mastin et al.
  (2009) relationship is still routinely used in VAAC operations and ensures that the ash forecasts are conservative.

- As expected with reduced emissions, the VATDM forecasts produced using the posterior emission ensemble with match-
  ing meteorology have ash clouds with much lower column loadings compared to the prior ensemble simulations, although
  they have a similar evolution. The simulations using the posterior emission ensemble have a much smaller range of col-
  umn loadings and are a closer match to the ORAC retrievals of ash column loading than the prior ensemble simulations.
  Thus, the Himawari observations constrain the ensemble spread.

- For this case study, the amount of ash emitted into the stratosphere is important. Higher fractions of ash (in terms of
  mass) are emitted into the stratosphere leading to higher peak stratospheric concentrations and ash plumes with greater
  horizontal extent when using the posterior ensemble. This could potentially increase the risk to aviation as this is near
  the cruise altitude of aircraft in the Pacific region.

- The risk-matrix approach to presenting ensemble forecast data has been applied to the VATDM simulations produced
  using the prior and posterior emissions from InTEM. In this case study, the use of the posterior emissions reduces the
  region of highest forecast risk by up to 51%. This has the potential to reduce disruption to civil flight plans. This result
  is consistent with that found in Harvey et al. (2020) and builds confidence in applying this methodology.

Future work will focus on applying this methodology to further case studies and comparing with ensemble and inversion
systems used by other modelling centres. Here the focus of the study was the impact of meteorological uncertainty on the
InTEM emission estimates and VATDM forecasts of ash location and the magnitude of ash column loadings but there are other
sources of uncertainty which could be incorporated into a full ensemble inversion scheme. These include uncertainties in ash
density and particle size and the representation of free tropospheric turbulence and wet deposition within the VATDM.



*Code and data availability.* The Himawari satellite data, NAME simulation and InTEM output are available in the University of Reading Research Data Archive at https://doi.org/10.17864/1947.000335. Further information about the data supporting these findings and requests for access to the data can be directed to n.j.harvey@reading.ac.uk. For InTEM and NAME licence enquiries, please contact the Met Office (atmospheric.dispersion@metoffice.gov.uk). The ORAC Himawari ash products can be obtained by contacting Andrew Prata (andrew.prata@physics.ox.ac.uk).

*Author contributions.* Conceptualization, N.J.H. and H.F.D.; methodology, N.J.H. and H.F.D.; software, N.J.H. and H.N.W.; validation, N.J.H., investigation, N.J.H.; resources, N.J.H., C.S. and A.P..; writing—original draft preparation, N.J.H.; writing—review and editing, N.J.H., H.F.D., H.N.W., C.S., A.P. and R.G.G.; visualization, N.J.H; funding acquisition, H.F.D. and R.G.G. All authors have read and agreed to the published version of the manuscript.

*Competing interests.* The authors declare no conflict of interest. The funders had no role in the design of the study; in the collection, analyses, or interpretation of data; in the writing of the manuscript, or in the decision to publish the results.

*Acknowledgements.* N.J.H., H.F.D., H.N.W. are funded by the Natural Environment research council (NERC) grant number NE/S005218/1. R.G.G. and A.P. are funded by NERC grant number NE/S003843/1. R.G.G. was also supported by the NERC Centre for Observation and Modelling of Earthquakes, Volcanoes, and Tectonics (COMET) and by NERC grant NE/S004025/1. We also thank Cathie Wells at the University of Reading for the providing the time optimal flight tracks used in Figure 8.





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
