# Peer review of "Quantifying the impact of meteorological uncertainty on emission estimates and risk to aviation using source inversion for the Raikoke 2019 eruption"

_Atmospheric Chemistry and Physics, 2021_

## Referee Comment (RC2)

**Referee comment for acp-2021-973, Quantifying the impact of meteorological uncertainty on emission estimates and volcanic ash forecasts of the Raikoke 2019 eruption by Natalie J. Harvey, Helen F. Dacre, Cameron Saint, Andrew Prata, Helen N. Webster, and Roy G. Grainger**

This study presents the results of using ensemble meteorology from the Met Office Global and Regional Ensemble Prediction System (MOGREPS-G) to drive dispersion simulations used in the Inversion Technique for Emissions Modelling (InTEM) algorithm. The authors use variables retrieved from Himawari satellite observations using the Met Office algorithm in the InTEM inversions. This results in a constrained eruption emission profile.

The success of the inversions are then assessed using variables retrieved from the same Himawari data, but using the independent Optimal Retrieval of Aerosol and Cloud (ORAC) algorithm. The results presented show how NAME dispersion simulations initialised using the constrained eruption emission profile show improved agreement with observation.

Finally, the authors show how the predicted risk to aircraft is reduced when the constrained emission profile is used in NAME simulations, and that disruption to air traffic could have been reduced.

The paper is well written and is highly suitable for publication in Atmospheric Chemistry and Physics after some clarifications and added explanations.

A slight concern I would like addressed regards the limitation of the satellite retrievals of mass column loadings. My understanding is that the mass retrievals are severely underestimated for pixels with high column loadings – which is likely to have been the case in the first 24hrs or so of the Raikoke eruption. Would it not be the case that missing this part of the ejected mass would result in the posterior mass profile being under estimated?

On a related note, it would be good to have some more discussion on how independent the two retrievals really are – they are after all using the same observations, so tuning the emission profile to get close to one set of satellite retrievals, and then using the other set to assess the success could be viewed as problematic. Some more discussion as to why this is ok please!

Please can the authors consider adding a table in the methods and data section giving a brief outline of each method, what they are used for, and some date ranges?

Minor points:

L66 Bent over plumes are discussed – would this reduce the mass estimate using the Mastin relationship?

L94 Brier skill score – please explain what this is

L127 – The MOGREPS met data is on a 20km grid resolution – my understanding is that this is less than the UM met data that is used in NAME operationally – please

comment on the effect of using lower resolution met data on the accuracy of the resulting dispersion simulations.

L158 – This is the first mention of InTEM – please define here.

L193 – Is the Hobbs et al. size distribution simply cut off, of has the shape been modified?

L161 – Do the Met Office and ORAC methods use met data? Do they both use the same? Is this ensemble data? Could the independence of the two satellite methods be improved by using ECMWF data for ORAC?

L257 and elsewhere – figure -> fig. I think ACP style asks for fig. unless at the start of a sentence – please check

L 263 – 265 I'm not sure what these sentences mean, please clarify.

L 269 – fig. 3(a) seems to show variation with height – I thought the prior was constant from vent to plume top?

Figure 5(a) and 6(a) – please explain what the grey pixels are

L325 - "In this case the cross section intersects the simulated ash plume in 3 locations" – it seems to me that the dashed line in fig. 6 a – b intersects the simulated plume almost entirely. Please clarify.

Figure 8 and discussion of same - The posterior emission profile is constrained using satellite data out until 00:00 25 June? L381 then claims that the disruption to air traffic could have been reduced on 22 and 23 June - Can the NAME simulation initialised using this profile then really be called forecasts as they could not have been produced prior to 25 June?

References – author names have been capitalised and I think some doi numbers are missing. Please check.

---

## Author Comment (AC1)

**Quantifying the impact of meteorological uncertainty on emission estimates and volcanic ash forecasts of the Raikoke 2019 eruption**

April 21, 2022

We thank the reviewers for their detailed reports. Below, the reviewer comments are in black and the responses in blue italics.

**1 Reviewer 1**

**Major comments**

1. The title "Quantifying the impact of meteorological uncertainty on emission estimates and volcanic ash forecasts of the Raikoke 2019 eruption" does only reflect parts of the work that has been presented in the manuscript. The forecast improvement due to the inversion derived emissions compared to the prior emissions estimate as well as the risk assessment for ash encounters of air traffic are significant contributions to this study but not represented in the title. Please consider adapting the title such that it reflects the whole manuscript.
*We have modified the title to reflect the inversion and risk aspects of the paper.*

2. Within the abstract, there is no hint on the model system used nor the satellite data sets being used for the inversion. The ensemble setup as well as the satellite data are an essential part of the study and should be named in the abstract. The same applies to the short summary of the aim of the paper (lines 32-40), where no tools are mentioned. Here, the essential information that the ensemble of ash dispersion simulations is based on perturbations in the meteorological forecast is missing.
*We have added information about the specific ensemble data, satellite data and dispersion simulations used in this study to the abstract. We have not changed the aim since it is to demonstrate the general method. I.e. the method we describe is applicable to any setup of ensemble meteorology, satellite retrieval and dispersion model.*

3. In line 101 to 103, the authors state that "ensembles can provide improved volcanic ash forecasts". However per definition, ensemble simulations describe a method where a

computer model is run a number of times with diverse perturbations. An improvement of volcanic ash forecasts cannot be expected by just applying ensemble forecasting. Please clarify this contradiction.

*We have changed the word 'improved' to 'useful additional forecast information' in the revised text.*

4. There are two different volcanic ash retrievals described and used in this study. However, the similarities and differences could be elaborated in more detail. When describing the ORAC retrieval method, the term of fine ash is firstly introduced in the manuscript (line 161). Is there a difference in the retrieved particle sizes compared to the Met Office algorithm? Please also elaborate if only the ORAC algorithm is able to retrieve volcanic ash properties over day and night (stated in line 167) or if it is also valid for the MetOffice retrieval. Overall, the purpose of using two Himawari-8 retrieved volcanic ash data sets could be described clearer from the beginning (MetOffice data used for the inversion and ORAC used for validation of inversion results). This is especially important when referring to the ORAC retrieval being independent satellite retrievals (line 289). Please elaborate to what extend they are independent, when building upon the same satellite, exploiting identical spectral range measurements, deriving the same retrieval quantity for identical time steps.

*We have added a short paragraph at the start of section 2.3 to clarify how the two different volcanic ash retrievals are used in the paper. Both the Met Office and ORAC retrievals can detect ash properties over day and night. Therefore, we have removed reference to time of day from the ORAC algorithm description. Both the Met Office and ORAC retrievals detect fine ash particles since they both use measurements detected by the AHI which cannot detect particles larger than 30 µm. Therefore, we have removed reference to fine ash from the ORAC algorithm description. As is already stated in the paper, the retrieval methods have been developed independently in different research groups, but they both use information from the Himawari-8 satellite. It is beyond the scope of the paper to perform a full intercomparison of the algorithms developed by these research groups. We have provided references to the papers describing them in the text.*

5. Even though the model setups and inversion method are successively introduced and explained with many details, it is hard to extract the main configurations used for performing the analysis being discussed in the results section. Therefore, there is an urgent need to include a paragraph explaining the different simulation time horizons, the corresponding eruption times as well as the time window of used retrieval data (with a clear distinction between retrieval data used for the inversion and for evaluation of results). What is the temporal resolution of the VATDM simulations? For example, the sentence extending from line 291-293 includes important information that should have been presented earlier.

*We have added a new figure 1, which clarifies the different simulation time horizons, eruption times and time window of retrieval data. The figure also distinguishes between data used for the inversion and evaluation of results. We have also renamed the results sections 4.1 and 4.2 to make it clear that there are two parts to the paper.*

**Minor comments**

1. In the abstract (line 18) and discussion (line 409), it is stated that the area deemed to be highest risk to aviation is reduced by 51 % comparing posterior to prior emission scenarios. However, in the section 5 this reduction is discussed to be 49 % (line 374).
   *Thank you for spotting this inconsistency. The area deemed to be highest risk to aviation is reduced by 49 %. We have corrected this in the abstract.*

2. As the uncertainties in eruption sources parameters and dispersion model parameters are not investigated within this study, their discussion takes a prominent amount of the abstract. I therefore suggest removing the sentence "The posterior inversion emission estimates are also sensitive to uncertainty in other eruption source parameters (e.g., the ash density and size distribution) and internal dispersion model parameters (e.g., parameters relating to the turbulence parameterisation).", and moving the sentence "Extending the ensemble inversion methodology to account for uncertainty in these parameters would give a more complete picture of the emission uncertainty, further increasing confidence in these estimates." to the end of the conclusion.
   *We included the sentence regarding other uncertainties, as the ACP guidelines state that abstracts should provide possible directions for prospective research. Therefore we have shortened the sentence but not removed it.*

3. In line 98, the authors state "similar agreement at short lead times". It remains unclear to what the similar agreement refers to. Is it an observation-model agreement or a similar agreement to results of more than 12 hours lead times? Further, the definition of "short lead times" would improve the discussion at this point.
   *Here 'similar agreement' refers to the accuracy of deterministic vs ensemble forecasts when compared to satellite observations. 'Short lead times' refers to lead times shorter than 12 hours. We have clarified this in the revised text.*

4. Please provide detailed references on MOGREPS-G (line 126). Is MOGREPS-G an ensemble generation system, which builds on a specific or any meteorological model, or is it a meteorological ensemble model system?
   *MOGREPS-G uses the Met Office Unified Model. We have moved the reference describing the details, Bowler et al. (2008), higher up this paragraph.*

5. The information provided in line 128 on the initialisation of forecasts 4 times per day and 7 days extent is probably related to operational forecasts. However, this is not relevant for this study, because only 12 UTC initialised forecasts are used. Instead, please provide more information on the meteorological simulations used: Are these meteorological forecasts or reanalysis? From which data are the forecasts initialised?
   *As stated in section 2.1, the MOGREPS-G forecasts used to drive the NAME simulations are initialized at 1200 UTC on 21 June 2019. Thus we think it is clear that we are using meteorological forecasts not reanalysis, and the date the forecasts are initialised. We provided more general information about MOGREPS-G for interested readers who might consider performing similar experiments with this meteorological data, so have not removed this information.*

6. In lines 143-144, the channels used to identify ash contaminated satellite pixels are discussed. Even the authors state that the retrieval algorithm is based on Frances et al. (2012), mentioning that the ash identification is based on the reverse absorption technique might be valuable. Please also explain if ash identification is applicable to pixels being dominated by meteorological clouds and how meteorological clouds are identified.
   *We have added the information that the Francis et al. (2012) method uses a reverse absorption technique. Three long-wave window channels are used to distinguish between meteorological cloud and ash clouds. We have added this information to the revised text.*

7. Please discuss if and how differently classified pixels (clear sky, unclassified, meteorological clouds, ash) are treated in the inversion algorithm. Are meteorological cloud pixels classified as unclassified or does unclassified relates the retrieval failures?
   *Pixels containing meteorological cloud are unclassified. The inversion algorithm only uses clear sky and ashy pixels. We have clarified this in the revised text.*

8. Please justify the selection of 50 % and 90 % in line 157. How do the 90 % build up of fractions containing ash or clear sky? How are for example ash column values calculated if 50 % of the pixels contain ash and 50 % of the pixels are unclassified?
   *These thresholds values are based on those developed by Pelley et al. (2021). The thresholds account for representativity errors, which arise when the ash and clear sky gridboxes are not representative of all pixels in a gridbox.*

9. Are the cloud-top pressures discussed in line 181 related to the troposphere/stratosphere model configuration? Then, the new paragraph should start one sentence earlier with "The troposphere/stratosphere model configurations are run by …". Otherwise, please explain how these cloud-top pressures relate to the 800 hPa cloud top in line 176.
   *The cloud-top pressures relate to the tropsophere/stratosphere model configurations. We have therefore changed the paragraph structure as suggested.*

10. Chapter 2 seems a bit mixed up. First describing all modelling related systems (MOGREPS-G and NAME), before introducing the retrieval algorithms and finally the inversion system would improve the guidance through methods and data.
    *We have reordered section 2 as suggested.*

11. Please explain if the prior source term (section 2.4.1) is also restricted to chosen model inversion resolution (4km height levels and 3-hourly time steps).
    *The prior source term is regridded onto the inversion resolution. This information has been added to the revised text.*

12. Are emissions or ash concentrations constrained to positive values (cf. line 227)?
    *The non-negative constraint is applied to the best estimate of emissions. This is stated in the text.*

13. It would be helpful to see the quadratic cost function (mentioned in line 229) to understand how emission profiles and dispersed volcanic ash states are contributing to the costs and what quantity is ultimately optimised by the inversion algorithm.

*The full description of the quadratic cost function is given in Pelley et al. (2021) (their equation 9). Rather than include this equation in the paper, we refer the readers to Pelley et al (2021).*

14. In section 3, the initial eruptive plume height of the Raikoke eruption is discussed. However, is there any information on the temporal evolution of the plume height? This could be also interesting with respect to section 4 – Results, where emission plume heights are discussed.
*As Raikoke is a remote island there are no regular high-time resolution observations of the plume height available, hence the need for alternative methods to estimate plume height. Our information about the plume heights comes from the Global Volcanism Program: Report on Raikoke. They report at least nine explosions (6 within the first 25 minutes) and a strong explosion at 1640 UTC on 22 June with an ash plume up to 10-11km. Since we have already referenced this source of information we have not altered the text.*

15. In figures 1 and 3, colormaps extend from 105 g/s to 108 g/s. Are emissions lower than 105 g/s visualised here? If not, please also include these emissions or explain where this lower limit of emissions comes from. Is this a limitation defined by the inversion algorithm?
*There are no posterior emissions below $10^5 g/s$. This is not imposed by the inversion algorithm.*

16. In lines 252-253, the authors claim, "Four members (e.g., member 13) have continuous emission of ash between 4-12 km above vent level (avl) whereas the other 14 members have times when there is no emission of ash at this height range.". However, ensemble member 13 is not part of the four members being characterized by continuous emissions between 4-12 km. And for the other 14 members, there are surely emissions at this height range, yet, not continuous.
*The reviewer is correct, member 13 has no ash between 4-12km avl in the last 6 hour window. Members 4, 8, 10 and 16 have continuous emissions between 4-12km. We have corrected this in the revised text.*

17. Please explain what is meant in line 263 by "There is also a range, 4459–5314, in the number of unique observations which impact the inversion between ensemble members.". What do the numbers refer to (a quantity of observations or a numbering of observations)? How does the ensemble meteorology sort out observations in the inversion process?
*To be used in the inversion, the NAME simulated and satellite observed pixels must either (i) both contain ash, or (ii) contain a clear sky flag in the satellite observations. This results in a range in the number of unique observations that are used for each ensemble since they have different dispersion patterns. We have re-written this sentence to provide more relative information. The largest number of observations used in the inversion process is 5314 for ensemble member 10, which is 17% larger than the smallest number used (4459 with ensemble member 7). This is due to the different dispersion patterns for each ensemble member.*

18. For all figures illustrating the horizontal dispersion of ash (fig. 5a-d, 6a-d, 7, and 8), it would be helpful to have some geographical orientation here. Please add longitude and latitude information to the figures.

    *We have added latitude and longitude labels to a panel in figures 5-8.*

19. Figure 5 and 6 need adjustments in their colormaps to fit the values mentioned in the discussion. The extreme values of the prior emission ensemble are not represented by the selected colormap limits. Further, it would be helpful to know, if the selected cross sections are following a selected line of longitude or if the cross sections are irregular due to the selected projection. What is the reason that some error bars of the Met Office algorithm extend to the x-axis (in panel e))? What is the lower limit of the y-axis?

    *Since concentrations typically decrease exponentially away from the plume axis, it is not possible to use a linear colourmap that shows both the extent of the ash cloud (defined by low concentrations) and also the peak ash concentrations, since they differ by over 2 orders of magnitude. Therefore we chose to use a linear scale for the maps (at the expense of saturation for peak concentrations) and a logarithmic scale for the vertical cross-sections. We think that this compromise allows us to compare both the horizontal extent of the ash clouds and the the quantitative column loading at the centre and edges of the plume.*

20. In lines 316-317, the authors describe that "Between 45-50 °N, the posterior ensemble spread encompasses the Met Office retrievals within their retrieval uncertainty." Isn't this an overfit of the inversion algorithm? Please discuss in more detail about the interpretation of this extreme fit, the choice of the prior emission error and the observation errors used for the inversion.

    *The error on the prior plume height is assumed to be +/- 2km which is then used to calculate the error in the mass eruption rate using the empirical relationship of Mastin et al. 2009. The observational errors are estimated from a combination of published errors in the AHI instrument for radiometric noise of the relevant HIMAWARI channels and errors in the forward radiative transfer model used. Thomson et al. 2017 contains a full description of how the prior and observational errors are used in the InTEM system. Since we have referenced Thomson et al. 2017 in section 2.4.3 we have not altered the text.*

21. "In this case the cross section intersects the simulated ash plume in 3 locations - 45-46°N, 49-51°N and 53-59°N." cannot be followed, as figure 6 reveals volcanic ash continuously extending between 43-60°N. Please clarify. Further use "–" in latex for the dash before the list of locations such it does not appear as a minus.

    *The reviewer is correct, the ensemble mean volcanic ash continuously extends between 43-60°N in the ensemble mean. However there are only 3 latitudes at which all members contain non-zero volcanic ash. We have clarified what we mean by the 3 locations. We have also replaced the "-" with "–" as suggested.*

22. It would be interesting to learn something about the detection limit of volcanic ash with Himawari retrievals, e.g., where discussing the detected ash of figures 5 and 6

(e.g. line 326).

*The detection limit for thermal infrared retrievals is 0.2 /gm$^2$ (Prata and Prata (2012). We have added this to the revised text.*

**Technical corrections**

1. Please make sure that all abbreviations used are introduced when used for the first time (e.g. line 131 – NAME, line 158 – InTEM) and that abbreviations are consistently used throughout the manuscript (i.e. in some cases, it just says VATD instead of VATDM).
   *InTEM is now introduced. We have reordered section 2 so NAME is already introduced. We now use VATDM throughout the paper.*

2. There are many sentences where commas would be appropriate to support readability.
   *We have added commas where we think they are appropriate.*

3. There is no subsection 2.2.1 needed since it is the only subsection to section 2.2.
   *We have removed subsection 2.2.1*

4. Please indicate, which two channels are used for the effective absorption optical depth ratio (line 148-149).
   *The channels used to calculate the optical depth ratio are the 12.4 µm and 10.4 µm channels. This has been added to the text.*

5. Please explain where the value of "10" is coming from in line 172.
   *We chose to threshold the retrievals using the cost function following the work of Thomas and Siddens (2015), who found that false detections were often associated with radiative transfer simulations that had a poor fit to the measurements. For the Raikoke case, we found that false detections (isolated pixels unrelated to the main volcanic plume/cloud) often had cost values higher than 10. Setting a cost threshold of 10 therefore provided a good balance between false positives and true positives. This explanation has been added to the text.*

   *Thomas, G. E. and Siddans, R.: Development of OCA type processors to volcanic ash detection and retrieval, https://doi.org/EUMETSAT RFQ 13/715490, 2015.*

6. Line 178: Please elaborate what "nearby" means at this point. Is it meant as nearby to the Raikoke volcano, nearby the ash cloud or something else?
   *Here nearby refers to nearby to the ash cloud.*

7. At this stage of Prata et al. in preparation cited in lines 184-185, please remove this sentence from the manuscript and add it again if the cited publication is published during the ongoing review process.
   *We have removed reference to the Prata et al. paper (in preparation) since the paper by McGarragh et al. also contains the relevant information.*

8. Are there any references on the algorithms used for "free tropospheric turbulence, sedimentation, dry deposition and wet deposition" (line 189)?

*The free tropospheric turbulence scheme is described in Webster et al. 2018. The sedimentation scheme and dry deposition scheme is described in Webster and Thomson 2011. The wet deposition scheme is described in Webster and Thomson 2017. These references have been added to the revised text.*

9. Is there any reference for the ash density of 2300 kg m-3 (line 190)?
   *A reference to Bonadonna and Phillips (2003) has been added to the revised paper.*

10. Please state what observations the plume height estimate relies on in line 208.
    *Bruckert et al. 2021 use satellite imagery from GOES-17 to estimate plume height. According to the Tokyo VAAC VAG they use satellite data, NWP data and pilot reports. We have added this information to the text.*

11. Please note that the release rates discussed in lines 219-220 refer to a first guess and are different to the prior source term.
    *We have reworded the sentence to explain how the prior source term relates to the first guess nominal release rate.*

12. Line 236: What does the "peak value" relate to? The ensemble distribution?
    *The peak value refers to the mode of the probability distribution. We have clarified this.*

13. Does 21-24 June refer to the ash retrievals used in the inversion? If so, this is in contradiction to section 2.4.3. Please clarify.
    *We are unsure why this sentence contradicts section 2.4.3. The inversion uses satellite retrievals from 21–24 June.*

14. Please check if the statement "especially below 4 km avl where ash is only emitted for one 3 hour period at the start of the eruption." is valid or if this finding is related to the choice of colormap having a lower limit at 105 g/s.
    *This statement is not dependent on the choice of colormap.*

15. Lines 273-274: There is no need of the "Note"-sentence in parentheses.
    *We have removed the parentheses and have shortened the sentence to provide the most relevant information.*

16. Figure 4: Is the fraction of total ash emitted in the three atmospheric levels dependent of the ensemble meteorology when applying the prior emission profile to ensemble simulations?
    *The aim of figure 4 is to show that the fraction of total ash emitted in the 3 atmospheric levels is dependent on the ensemble meteorology. The final bar shows the prior emission profile. We are unsure what the reviewer means by applying the prior emission profile to the ensemble simulations as this would not change the emissions profile, only the atmospheric concentrations.*

17. Line 280: Please change "other parts" to "lower regions".
    *Changed*

18. Line 281: Please state that de Leeuw et al. (2020) also relates to the Raikoke eruption.
*This information has been added.*

19. In lines 304-305, the authors state to show a difference in figure 6. However, the figure does not contain difference plots. Further, there is no reference to the illustrated prior ash cloud. Please rephrase this sentence to make clear what is shown in figure 6.
*We have reworded this sentence to make clear what is shown in figures 6(a) and (b).*

20. Please explain what "matching ensemble" means (e.g. caption fig. 5 or line 332).
*Matching ensemble refers to the process by which both the inversion and the forecast use the same meteorology. We have clarified this.*

21. In line 315, it is stated that "Between 46.5-48.5° N, the posterior ensemble spread falls within the uncertainty of the ORAC retrievals.". Contrarily, figure 5 shows an ORAC retrieved column loading at 48 °N, which does not match the posterior ensemble spread.
*We agree that the posterior ensemble spread only falls within the uncertainty of the ORAC retrievals between 46.5–47.6° N and 48.4–48.7° N. The ORAC retrieval at 48° N is outside the spread. We have corrected this.*

22. Line 318: Please state what "mean magnitude" is meant here. Is it the magnitude of the cross-section evolution ensemble mean volcanic ash column mass loading?
*Mean magnitude refers to the ensemble mean ash column loading. We have clarified this.*

23. Please be more precise when discussion latitude ranges of figure 6e. In line 326, Himawari retrievals used in the inversion (thus MetOffice retrievals only) detect ash between 49-52°N.
*Thankyou for spotting this error. We have corrected 47°N to 49°N.*

24. Please add "by comparing the different ensemble members" to the sentence "At this time, the ash plume structure at all three levels is qualitatively very similar." (in line 338).
*This wording has been added.*

25. Line 337: Please correct "the maximum over" to "the maximum of".
*Corrected*

26. Please clarify what fraction of ensemble members is used in line 360. Do you use the ensemble mean, the ensemble minimum or does the fraction consists of selected ensemble members?
*To clarify this sentence and to be consistent with Prata et al. (2019) we have changed 'fraction of ensemble members exceeding low, medium and high ash concentrations' to 'likelihood of ash concentrations exceeding different concentration thresholds'.*

27. Line 396: Please connect "over estimated" to "overestimated".
*Corrected*

**2  Reviewer 2**

**Major corrections**

1. A slight concern I would like addressed regards the limitation of the satellite retrievals of mass column loadings. My understanding is that the mass retrievals are severely underestimated for pixels with high column loadings – which is likely to have been the case in the first 24hrs or so of the Raikoke eruption. Would it not be the case that missing this part of the ejected mass would result in the posterior mass profile being under estimated?
   *The total fine ash mass may have been underestimated during the first 12 hours due to opaque regions in the centre of the plume at the beginning of the eruption. However, once the plume evolved and became semi-transparent, we were able to retrieve the mass loading (via the optical depth and effective radius) and get a reliable estimate of the total fine ash mass. Since we are only trying to model the fine ash emission profile, and the satellite retrievals are sensitive to these particles (effective radii from 0.1–15 micron), it's not obvious that we should expect the source term to be underestimated.*

2. On a related note, it would be good to have some more discussion on how independent the two retrievals really are – they are after all using the same observations, so tuning the emission profile to get close to one set of satellite retrievals, and then using the other set to assess the success could be viewed as problematic. Some more discussion as to why this is ok please!
   *The reviewer is correct that both the Met Office and ORAC retrieval algorithms use data from the same satellite instrument (AHI). Therefore, as described in response to the comment above, any bias in the data will likely be reflected in both retrievals. However, there are currently no alternative high time-resolution satellite retrievals with which we can validate the observation constrained NAME forecasts independently. The main aim of the paper is to quantify the impact of uncertainty on emission estimates which is answered without the validation step. The validation step however, allows us to demonstrate that our results would be insensitive to the choice of satellite retrieval. We have added this discussion to the conclusions section.*

3. Please can the authors consider adding a table in the methods and data section giving a brief outline of each method, what they are used for, and some date ranges?
   *We have included a new figure 1 which shows the data used and their date ranges*

**Minor corrections**

1. L66 Bent over plumes are discussed – would this reduce the mass estimate using the Mastin relationship?
   *Bent over plumes would result in a lower estimate of the plume height, thus reducing the mass estimate using the Mastin relationship. However, the Raikoke eruption was explosive, with a plume penetrating into the stratosphere. Therefore, the effect of wind on the plume is likely to be small.*

2. L94 Brier skill score – please explain what this is

   *This sentence refers to published work by other authors and is not the focus of the current paper we have changed 'Brier skill score' to 'skill'. This keeps the meaning of the sentence but avoids a distracting explanation of the exact skill measure used.*

3. L127 – The MOGREPS met data is on a 20km grid resolution – my understanding is that this is less than the UM met data that is used in NAME operationally – please comment on the effect of using lower resolution met data on the accuracy of the resulting dispersion simulations.

   *The met data used to run NAME operationally is taken from the Global configuration of the Unified Model (UM). The horizontal resolution of the current global configuration increased to 10 km in 2017. Using lower resolution data results in less accurate dispersion simulations as mesoscale features are not resolved. However, it has been shown that ensembles of lower-resolution models can provide greater skill than single forecasts of higher-resolution models, particularly over time periods greater than 12 hours. In addition, ensembles are inherently probabilistic so can express uncertainty directly. We have added a discussion of the effect of using low resolution ensembles to the text.*

4. L158 – This is the first mention of InTEM – please define here

   *InTEM has been defined.*

5. L193 – Is the Hobbs et al. size distribution simply cut off, of has the shape been modified?

   *The Hobbs et al. (1991) measurements were airborne in situ and remote sensing measurements of particles $< 48\mu m$ in diameter. The NAME default particle size distribution ranges from $0.1$–$100\mu m$, therefore none of the Hobbs et al. size distribution has been cut off.*

6. L161 – Do the Met Office and ORAC methods use met data? Do they both use the same? Is this ensemble data? Could the independence of the two satellite methods be improved by using ECMWF data for ORAC?

   *The satellite retrievals do not use meteorological data.*

7. L257 and elsewhere – figure - fig. I think ACP style asks for fig. unless at the start of a sentence – please check

   *The reviewer is correct. According to ACP guidelines 'The abbreviation "Fig." should be used when it appears in running text and should be followed by a number unless it comes at the beginning of a sentence'. This has been changed.*

8. L 263 – 265 I'm not sure what these sentences mean, please clarify.

   *There is a range of 17% in the number of observations used in the inversion process for different ensemble members due to the variation in dispersion patterns. We have clarified this in the revised text.*

9. L 269 – fig. 3(a) seems to show variation with height – I thought the prior was constant from vent to plume top?

   *The prior height is 13km avl. This emission profile is regridded on to the height-time*

*grid used in teh inversion. As a result, the emissions are constant from the vent top to 12 km avl and the remaining emission between 12–13 km avl is redistributed between 12–16 km avl, hence the reduction in emissions in this layer. We have explained this in the text.*

10. Figure 5(a) and 6(a) – please explain what the grey pixels are

    *The grey shading indicates grid boxes that are classified as clear sky. This information is already included in the figure 5 caption.*

11. L325 – "In this case the cross section intersects the simulated ash plume in 3 locations" – it seems to me that the dashed line in fig. 6 a – b intersects the simulated plume almost entirely. Please clarify.

    *The reviewer is correct, the ensemble mean volcanic ash continuously extends between 43-60°N in the ensemble mean. However there are only 3 latitudes at which all members contain non-zero volcanic ash. We have clarified what we mean by the 3 locations.*

12. Figure 8 and discussion of same - The posterior emission profile is constrained using satellite data out until 00:00 25 June? L381 then claims that the disruption to air traffic could have been reduced on 22 and 23 June - Can the NAME simulation initialised using this profile then really be called forecasts as they could not have been produced prior to 25 June?

    *We refer to the NAME simulations as forecasts since they are free running simulations initialised at 18UTC on 21 June, using forecast ensemble meteorology. The posterior emissions used in the NAME simulations does use data from a time window beyond the validation times (see figure 1). However, this does not prevent us from achieving our primary goal of using the posterior ensemble spread to quantify the impact of uncertainty in the inversion estimate of ash emissions. We have made this clearer in the revised text.*

13. References – author names have been capitalised and I think some doi numbers are missing. Please check.

    *The capitalisation has been corrected. We have provided doi numbers for all of the references we can find doi for.*